# Climate Literacy—Imperative Competencies for Tomorrow's Engineers

**Susan E. Powers** [1,*], **Jan E. DeWaters** [2] **and Suresh Dhaniyala** [3]

1 Institute for a Sustainable Environment, Clarkson University, Potsdam, NY 13699, USA
2 Institute of STEM Ed, Clarkson University, Potsdam, NY 13699, USA; jdewater@clarkson.edu
3 Department of Mechanical and Aeronautical Engineering, Clarkson University, Potsdam, NY 13699, USA; sdhaniya@clarkson.edu
* Correspondence: sep@clarkson.edu or spowers@clarkson.edu; Tel.: +1-315-268-6542

**Abstract:** Engineers must take a leading role in addressing the challenges of mitigating climate change and adapting to the inevitable changes that our world is facing. To improve climate literacy, technical education must include problem formulation and solutions that consider complex interactions between engineered, Earth, and societal systems, including trade-offs among benefits, costs, and risks. Improving engineering students' climate literacy must also inspire students' motivation to work toward climate solutions. This paper highlights the content and pedagogical approach used in a class for engineering students that helped contribute to significant gains in engineering students' climate literacy and critical thinking competencies. A total of 89 students fully participated in a pre/post climate literacy questionnaire over four years of study. As a whole, students demonstrated significant gains in climate-related content knowledge, affect, and behavior. Substantial differences were observed between students in different engineering disciplines and male vs. female students. Assessment of critical thinking showed that students did an excellent job formulating problem statements and solutions in a manner that incorporated a multidimensional systems perspective. These skills are critical for students to address climate change effectively in their eventual professions.

**Keywords:** climate; project-based learning; climate literacy assessment; engineering education



## 1. Introduction

The engineering profession must take a leading role in addressing the challenges of mitigating climate change and adapting to the inevitable changes that our world is facing [1]. Smith et al. [2] suggest that grave challenges, including renewable energy sources and climate change, require ambitious engineering innovation from a diverse pool of creative engineering talent. They also suggest that the engineering profession must assume a pivotal leadership role by taking a seat on the national policy table to help shape efforts to address grand challenges such as climate change.

Professional engineering societies and accreditation bodies reinforce the expectation for future engineers to tackle grand challenges. The World Federation of Engineering Organizations (WFEO) noted that it is critical to equip engineering graduates from all disciplines with the relevant knowledge and skills to effectively address society's sustainable development challenges such as climate change. In the United States, the Accreditation Board for Engineering and Technology (ABET) requires engineers to be able to:

(1) apply engineering design to produce solutions that meet specified needs with consideration of public health, safety, and welfare, as well as global, cultural, social, environmental, and economic factors, and

(2) recognize ethical and professional responsibilities in engineering situations and make informed judgements, which must consider the impact of engineering solutions in global, economic, environmental, and societal contexts (Criteria 3.2 and 3.4) [3].

The incorporation of engineering for sustainable development (ESD) concepts in engineering education is also supported by discipline-specific documents such as the American Society for Civil Engineers' code of ethics, which includes sustainability in the tenets of practice [4].

Many first-year engineering students do not believe in anthropogenic causes of climate change. For example, fewer than half of the engineering-bound students surveyed by Shealy et al. [5,6] strongly agreed or agreed that climate change is caused by humans. The students who had prior exposure to climate change science in high school were more likely than their counterparts to be interested in using their engineering skills to address climate change and other global sustainability problems.

Engineering students in the U.S. receive very limited exposure to climate science in their undergraduate education [7]. As stated by Fri [1], technical education often does not focus on the multiple, complex interactions between engineered systems and the Earth's climate system, nor does it recognize that transformation raises societal challenges, including trade-offs among benefits, costs, and risks.

There is a tremendous opportunity for engineering students and practicing engineering professionals to contribute to mitigation and adaptation solutions. Improving students' climate science *literacy* will require educational strategies that not only strive to increase climate- and energy-related content knowledge but also improve their ability to solve energy- and climate-related problems and inspire and motivate them to work toward solutions.

Climate change as part of engineering education can provide relevance and **context** for the broader skills needed for today's engineering graduates [8–11]. Situating the development of technical skills, which are typical of engineering classes, within projects that require students to address complex, wicked problems can better prepare them for careers that demand an understanding of the socio-technical nature of engineering [12]. Classes organized around the problems and solutions for addressing climate change can also meet the needs of engineering education across disciplinary boundaries.

Based on a parallel review of education research related to climate change education (CCE), engineering for sustainable development (ESD), and engineering education, common themes emerge that illustrate the relevance of climate change education in meeting many of the needs of engineering students. The three strands that define literacy—(1) knowledge, skills, and competencies; (2) attitudes (affect); and (3) behaviors—are all relevant for engineering in general and engineering for climate change. The overlap between engineering education and CCE illustrates that teaching engineering within the context of solving sustainability problems such as climate change can help to create engineers who are fully aware of what is going on in society and have the skills to deal with the societal aspects of technologies (e.g., [13–15]). Critical among these skills is an awareness and respect for environmental and social issues at both local and global scales, transdisciplinary (or multidimensional) and systems thinking to move beyond defining solutions based only on technology and cost criteria, and an understanding of the complexities of our global grand challenges. Critical needs for CCE that are not typically expected in engineering education are the attitudes and behaviors that stress the personal and professional motivation to act.

From a pedagogy standpoint, active learning is crucial in any class that includes ESD or climate change topics (e.g., [16]). Although lecturing is an effective means of teaching some basic climate science, other pedagogies including project- or problem-based learning, case studies, and role play are examples of strategies required to promote the broader systems and critical-thinking skills needed for climate literacy [13,17]. This need to diversify teaching methods is consistent with a wide range of engineering education literature that confirms the value of engaging students in active learning and problem solving to enhance critical-thinking skills, increase long-term retention of information, and increase student retention in STEM majors (e.g., [18–21]).

## 2. Project Overview and Objectives

The authors of this paper created a climate science course for engineers at Clarkson University in 2010 (Potsdam, NY, USA). Its primary objectives include providing the necessary background to (1) permit students to demonstrate an understanding of the workings of the Earth's climate system and anthropogenic activities that are altering these processes and (2) apply their engineering skills to assess the efficacy of mitigation measures or adaptation strategies to address the climate crisis. Interactive class lectures, projects, and assignments support students' need to learn the science, causes, current global policies, and connections between engineered systems, social systems, and climate change. A semester project engages them in applying their engineering skills to address specific climate change mitigation or adaptation strategies.

The premise that climate literacy education programs should address both content knowledge and competencies to address climate change problems provides a basis for this class. Effective instruction must incorporate scientifically based knowledge and observations and foster critical-thinking, problem-solving, and decision-making skills so that students are better able to apply knowledge and skills in confronting and analyzing new, unfamiliar situations. Accessing and analyzing real earth science and energy system data and model projections are fundamental to the inquiry and project-based instructional modules. This use of real-world data facilitates a focus on the science, mathematics, and engineering applications; the exploration of questions related to the causes and impacts of climate change; and the nature of policy or engineering interventions to mitigate or adapt to these changes.

The objective of the research presented in this paper was to quantify and interpret gains in climate literacy and associated self-efficacy, problem-solving, and critical-thinking competencies made by engineering students in this class. The quasi-experimental study examines the overall status of climate literacy and evaluates changes in students' climate-related content knowledge, affect, and behavioral attributes following completion of the semester course. The specific questions that the research aims to answer include:

1. How climate literate are engineering students?
2. Does this course increase their overall climate literacy?
3. At the end of the semester, do the students have the self-efficacy to effectively integrate climate change perspectives into their profession?
4. At the end of the course, have the students demonstrated that they have the technical and critical-thinking skills and competencies to address climate change in their profession?

These questions were explored with a variety of assessment instruments. Variables evaluated in this assessment included the students' gender, year that they took the class, grade level, prior exposure to climate change in a different class, and major. Based on prior experiences and general knowledge about females in engineering majors, we expected that females and students in environmental and civil engineering majors to be more climate literate at the start of the class due to their self-selected bias towards choosing a discipline that directly serves society and the environment. The study used only one pedagogical approach based on project experiences and a high level of critical thinking. We were not able to assess how other types of classes would impact engineering students' climate literacy.

## 3. Methods

### 3.1. The Course—Global Climate Change: Science, Engineering, and Policy

The climate change class was developed primarily for undergraduate students from any engineering discipline. It has been taught every year or two since 2010 based on faculty workload and availability, but consistently has more students interested in taking the class than seats available. Due to the rapid changes associated with this topic, it is taught with web-based resources instead of a textbook. Table 1 identifies the units and some of the key resources used to provide active pedagogical approaches. With the rapid advances in

climate science, the resources available each year expand at an amazing rate. Resources presented in Table 1 are reflective of the materials available in 2021.

**Table 1.** Climate change class units and key resources for active learning.

---

***Our Climate is Changing (1.5 weeks)***

Historical evidence of climate changes, temperature proxies, and local and global impact

*EPA Climate Change Indicators (Available online: https://www.epa.gov/climate-indicators (accessed on 6 January 2021))*
*NOAA State of the Climate report (Available online: https://www.ncdc.noaa.gov/sotc/ (accessed on 18 January 2021))*
*Long term weather station monthly temperature data access (NASA GISS Available online: https://data.giss.nasa.gov/gistemp/(accessed on 18 January 2021))*

Focus on ice and the Arctic

*Access climate data from satellite measurements (MyNASAData Earth System Data Explorer Available online: https://mynasadata.larc.nasa.gov/EarthSystemLAS/UI.vm (accessed on 2 February 2019))*
*Glacier science and calving (Available online: https://chasingice.com/ (accessed on 15 January 2021))*
*Arctic sea ice coverage visualization and data (National Snow and Ice Data Center Available online: http://nsidc.org/soac/sea-ice.html#seaice (accessed on 15 January 2021); NASA Global Climate Change Available online: https://climate.nasa.gov/vital-signs/arctic-sea-ice/ (accessed on 15 January 2021))*

---

***Climate Science (4 weeks)***

Basic science—black body radiation, absorption of energy by greenhouse gases (GHGs) as a function of wavelength and molecular structure

*Greenhouse effect graphical simulator (Available online: https://phet.colorado.edu/en/simulation/legacy/greenhouse (accessed on 25 January 2021))*
*Molecules and light → GH gases (Available online: https://phet.colorado.edu/en/simulation/molecules-and-light (accessed on 25 January 2021))*

Carbon cycle—sources and sinks of GHGs, trends in GHG concentration, GHG inventories

*Carbon cycle (Available online: http://www.globalcarbonatlas.org/en/outreach (accessed on 1 February 2021))*
*US mix of electricity generation resources and GHG emissions (Available online: https://www.epa.gov/energy/power-profiler#/ accessed on 15 February 2021)*
*US GHG Inventory (Available online: https://www.epa.gov/ghgemissions/inventory-us-greenhouse-gas-emissions-and-sinks (accessed on 15 February 2021))*
*Global Carbon emission (Available online: http://www.globalcarbonatlas.org/en/CO2-emissions; Available online: https://ourworldindata.org/co2-and-other-greenhouse-gas-emissions (accessed on 22 February 2021))*
*Carbon dioxide concentrations (Scripps CO2 program Available online: https://scrippsco2.ucsd.edu/data/atmospheric_co2/index.html (accessed on 1 February 2021))*

Modeling and Predicting Future Climate (1.5 weeks)

*Science of climate modeling (Available online: https://www.climate.gov/maps-data/primer/climate-models (accessed on 8 March 2021))*
*Social science aspects of climate modeling—scenarios for future socio-economic futures*
*IPCC Emission scenarios (Available online: https://www.ipcc.ch/report/emissions-scenarios/ (accessed on 15 March 2021))*
*Shared socioeconomic pathways (Available online: https://www.carbonbrief.org/explainer-how-shared-socioeconomic-pathways-explore-future-climate-change (accessed on 15 March 2021))*

Access to and interpretation of climate model results

*Access IPCC results of model predictions (IPCC, Data Distribution Center, Available online: http://apps.ipcc-data.org/maps/ (accessed on 1 March 2019))*
*Climate Explorer: Climate metrics—historical and projected by zipcode (Available online: https://crt-climate-explorer.nemac.org/ (accessed on 8 March 2021))*
*En-ROADS global energy use simulation (MIT Climate Interactive Available online: https://www.climateinteractive.org/tools/en-roads/ (accessed on 12 April 2021))*

---

**Table 1.** *Cont.*

| |
|---|
| ***Mitigating Climate Change (2.5 weeks)*** |
| Mitigation strategies and technologies |
| *Drawdown (Available online: https://drawdown.org/solutions/table-of-solutions (accessed on 29 March 2021))*<br>*IPCC—1.5 deg. Futures (Available online: https://www.ipcc.ch/sr15/chapter/spm/ (accessed on 15 March 2021))*<br>*NAS—Negative Emissions Technologies (Available online: https://www.nap.edu/resource/25259/Negative%20Emissions%20Technologies.pdf (accessed on 5 April 2021))*<br>*IEA—Net Zero by 2050 (Available online: https://www.iea.org/reports/net-zero-by-2050 (accessed on 5 April 2021))*<br>*En-ROADS global energy use simulation (MIT Climate Interactive Available online: https://www.climateinteractive.org/tools/en-roads/ (accessed on 12 April 2021))* |
| International, national, and state-level policy mechanisms |
| *UN Climate Change—Paris Accord NDCs (Available online: https://unfccc.int/process-and-meetings/the-paris-agreement/nationally-determined-contributions-ndcs/nationally-determined-contributions-ndcs (accessed on 29 March 2021))*<br>*New York State Climate Law (Available online: https://climate.ny.gov/ (accessed on 29 March 2021))* |
| ***Adapting to Climate Change (1.5 weeks)*** |
| *Adaptation and Resilience (US Climate Resilience Toolkit Available online: https://toolkit.climate.gov/#explore (accessed on 19 April 2021))*<br>*Sea Level rise (NOAA Sea Level Rise viewer Available online: https://coast.noaa.gov/digitalcoast/tools/slr.html (accessed on 19 April 2021))* |
| ***Synthesis (2 weeks)*** |
| *International climate treaty negotiation game (MIT Climate Interactives, Available online: https://www.climateinteractive.org/tools/world-climate/ (accessed on 20 April 2019))*<br>*Project presentations* |

This class was designed with the specific intent to delve into the science of climate changes, consequences, and solutions. An important attribute of the class included the use of peer-reviewed and widely accepted scientific evidence, including the skills to identify reputable resources. Public discourse arising from the politicization and polarization surrounding this subject was not a focus of the class materials or discussion other than to acknowledge that it has affected policies and efforts to address the problem.

The semester project is the most significant component of the course grade. Weekly deliverables, assigned in a just-in-time manner as the students learn modeling tools and engineering or social strategies for mitigation and/or adaptation, help keep the students on track toward project completion. Students chose their own project statement and scope of work given the assignment requirements and grading rubric. Aspects of the project assignment included in Figure 1 illustrate the nature of the quantitative and critical thinking required from the students as they answer questions related to the causes, consequences, and solutions to mitigate or adapt to a changing climate. The full assignment is available upon request.

They addressed topics as diverse as the value of large-scale societal conversion to a vegetarian diet to the potential ability of increased nuclear power production in the U.S. to reduce GHG emissions. Topics related to climate science and adaptation were also among the completed projects.

The goal of the semester project is to address a research question related to climate change using a variety of databases and quantitative analysis. The project requires the use of real-world data from NASA, NOAA, DOE, IPCC or other agencies and critical analysis of the causes and consequences of climate change, and/or decisions necessary for mitigating or adapting to these changes. This project is NOT a term paper. It requires quantitative analysis.

1. Select a research question: Your research question will necessarily focus on only a narrow component related to these questions. Your research question should:
    cover at least two steps in this cause-effect chain;
    be derived from news or scientific literature regarding climate change;
    require the use of real-world quantitative data AND additional literature/data resources;
    include some level of quantitative analysis to evaluate and formulate an answer to your question; and,
    include some aspect of prediction of future changes

2. Develop a research plan to answer the question
3. Complete your analysis and communicate your results.

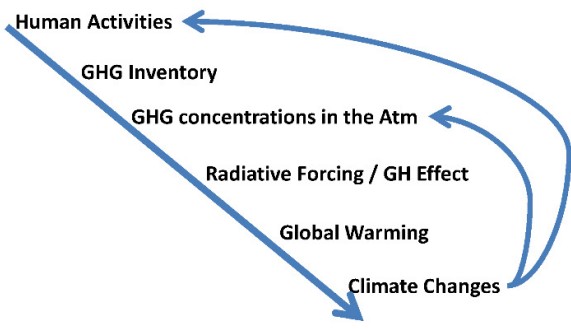

Cause and effect relationships in anthropogenic and natural systems that affect climate and social systems.

**Figure 1.** Selected details from the semester project assignment illustrate expectations for the project.

### 3.2. Participants

The climate change class is an elective for all students enrolled. It is one of the core course options for environmental engineering students and one of three environmental impact class choices for students enrolled in a sustainable energy systems engineering minor.

A total of 109 undergraduate students enrolled in the course over the four semesters (five-year duration) were evaluated for this research. The students were primarily juniors (30%) and seniors (61%) from a range of engineering disciplines. Most students (41%) were from mechanical and aeronautical engineering (MAE), though this comprised only 5% of the total number of junior or senior AE and 11% of ME students. Other participating disciplines included environmental (EnvE) (27% of the students in the class/44% of EnvE students took the class)), chemical (ChE) (8%/7%), civil (CE) (6%/5%), and electrical (EE) (4%/4%) engineering, as well as students from a cross-disciplinary program in engineering and management (E&M) (12%/8%) (Figure 2). The high percentage of mechanical engineers is consistent with their very high predominance in the School of Engineering (37% of all engineers) and their interest in sustainable energy systems. Females represent 26% of the course participants, which is substantially higher than the 17–21% value for the entire School of Engineering over the equivalent period.

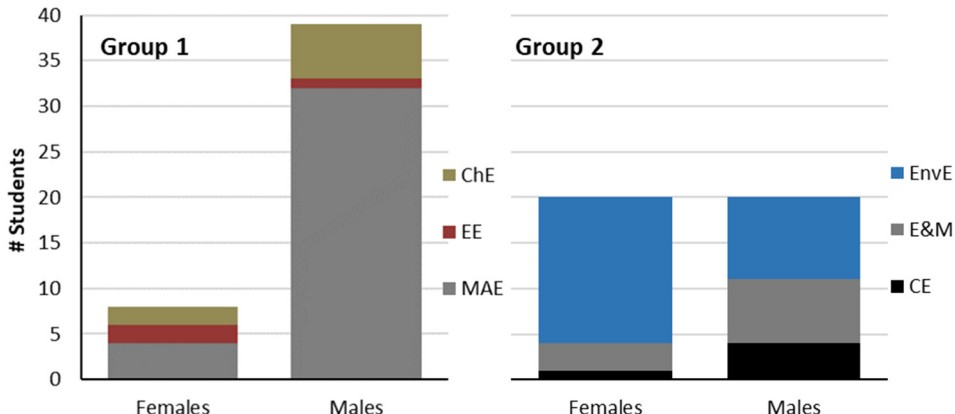

**Figure 2.** Distribution of participants by gender and major.

To evaluate the relationship between students' major and their climate literacy, students were aggregated by major into two groups due to low numbers in some disciplines. Group (1) included MAE, EE, and ChE students (n = 47), for which the ABET program criteria are highly technical in nature. Group (2) included CE, EnvE, and E&M (n = 41), for which program criteria include broader expectations such as multidimensional systems

thinking, policy, leadership, and/or sustainability principles. Group 1 is dominated by male MAE students, whereas Group 2 is dominated by EnvE students with an overall equal distribution between male and female students. The gender difference among majors was expected. For example, among 2018 B.S. engineering graduates in the United States, the percentage of females was 50.6% in environmental engineering and only 14.8% in mechanical engineering [22]. It was expected that students in Group 2 would be more climate literate, even before the class. (Note: only 88 students are included in this comparison; one of the students who completed the pre-post survey did not specify their engineering discipline).

### 3.3. Survey Instruments and Implementation

Two primary tools used a mixed-methods approach to assess student outcomes and address the research questions presented above. Existing survey tools were used to the extent possible to ensure the reliability of the methods. A *Climate Literacy Questionnaire*, previously described by Powers et al. [23] was used to measure students' climate literacy. Broader competency and professional attitudes were assessed with a *Climate Change and Critical Thinking Skills and Competencies Rubric* that was applied to semester project reports.

The *Climate Literacy Questionnaire*, developed as part of this research, provided a quantitative measure of the change in targeted content and personal competencies and attitudes toward global climate change among the students who participated in the course. The *Climate Literacy Questionnaire* was developed according to established psychometric principles and methodologies in the sociological and educational sciences (e.g., [24–26]). The instrument's content objectives were guided primarily by the 2009 document, *Climate Literacy: The Essential Principles of Climate Sciences* [27]. The intent was to understand how the course impacted students' broad energy literacy, rather than their retention of key course concepts. The questionnaire contains three subscales that measure climate-related affective (16 items), behavioral (13 items), and cognitive aspects (42 items), with four self-efficacy items embedded within the affective subscale and two additional questions that request that students provide a self-assessment of their knowledge about global climate change and their energy conservation behavior. An additional question about their self-efficacy to solve climate problems as an engineer was included in the post-survey only. The affective, behavioral, and self-assessment items use a 5-part Likert-type response scale with one neutral response; items in the cognitive subscale use a 5-option multiple-choice format. Topics in the cognitive subscale include climate science and the greenhouse gas effect, causes and effects of climate change, and potential mitigation strategies. A complete copy of the instrument and results is available.

The *Climate Literacy Questionnaire* was administered electronically in class at the beginning (pre) and the end (post) of the semester-long course. Pre/post responses were matched according to anonymous codes. Likert-type ratings were converted to numerical values according to a predetermined preferred direction of response. Several different nonparametric statistical procedures were used to investigate pre/post changes in student performance on each of the three climate literacy subscales, as well as to explore potential differences in student outcomes among various student cohorts. Student gains within each subscale were calculated using a Wilcoxon signed-rank test, the nonparametric equivalent of a paired t-test. Between-group comparisons were determined with a Mann–Whitney test for independent samples.

Students' climate change problem-solving and critical thinking competencies were assessed with a rubric, adapted here from related work [28]. The rubric, which measures four primary attributes (Table 2), recognizes that critical-thinking skills for engineers require both (1) a logical process for problem formulation through solution and (2) questioning the validity and quality of the data and analysis in each step in that process [29]. A 5-point scale was established for each of the attributes ranging from 1 (in progress) to 5 (superior). Although this rubric does not explicitly address climate change, the project assignment (Figure 1) did identify explicitly how the projects needed to assess critically an aspect of climate science or solutions. The *Competencies Rubric* was applied by co-authors to 30

semester reports at the end of the term. Inter-rater reliability was assessed and determined to be acceptable according to standard methods [30].

**Table 2.** Climate change and critical-thinking skills and competency rubric criteria.

| Rubric Attributes | Competency Expectations—Example for a "Proficient" Score of 4 |
|---|---|
| Formulates problem/question or issue | • Satisfactorily identifies and clarifies problem<br>• Describes within the context of the broader issue<br>• Recognizes key points or issues among details in relation to given question. |
| Uses data and evidence appropriately, objectively, and systematically to address a problem | • Approach to and use of data/evidence is organized<br>• Some assumptions stated<br>• Examines quality of data and other source of evidence |
| Formulates evidence-based conclusion or problem solution | • Applies evidence-based interpretation of data to solution of the problem<br>• States conclusion or problem solution, shows how conclusions or solutions emerge from the evidence or data<br>• Demonstrates relationship to the question, with context of larger implications |
| Evaluates solution | • Assesses solution in terms of its reliability and its need for further evidence.<br>• Assesses implications of solutions to specific questions or problems in context of larger issue<br>• Evaluates trade-offs, benefits, and detriments of various solutions. |

## 4. Results and Discussion

### 4.1. Climate Literacy

Before taking the climate change class, students had low scores on all three climate literacy sub-scales (Figure 2), with a mean of 62.4% correct on the knowledge scale and only 20% of the students achieving a knowledge "passing" rate of 70%. This is consistent with the generally low climate literacy among college students found by others (e.g., [31]). Mean scores on the affect and behavior scores were also below the target 3.8 score (equivalent to 70%). Our students share the common misconceptions about the role of the ozone depletion and toxic chemicals contributing to climate change (Table 3). Huxster et al. [31] attribute these common misconceptions as a general "pollution conceptual model", where students lump all environmental impacts into one category and believe that any "good environmental practice" will help to mitigate climate changes. This confusion and aggregation can lead to misplaced priorities and efforts when trying to tackle the specific causes and impacts associated with climate changes. Gautier et al. [32], who studied misconceptions about 21 key principles related to the greenhouse effect, suggest that instructional strategies that require students to explicitly address and evaluate their ideas, through role play, discussion/debate, or even presentation of information, will help them correct their misconceptions and allow for learning more scientific conceptions.

As a whole, the post-results show that students demonstrated significant gains in climate-related literacy (Figure 3). Of the 89 students who fully participated in the pre/post questionnaire, there were statistically significant gains in content knowledge ($p \ll 0.001$), affect ($p \ll 0.001$), and behavior ($p = 0.002$). Mean post scores were at or above a 'passing' cutoff ($\geq$70% of the maximum knowledge score; >3.8 for affect and behavior), with 68.5% of the students achieving the knowledge target of $\geq$70% correct. This showed substantial improvement over the pre-scores, for which none of the three subscale means met the targeted passing score.

**Table 3.** Student gains on selected cognitive items.

| Survey Items | Pre-Test Average | Post-Test Average [1] |
|---|:---:|:---:|
| **Climate Science** | | |
| $CO_2$ is the greenhouse gas we are most concerned about limiting emissions of, to reduce global warming. | 74.2 | **95.5 \*\*\*** |
| Energy in the infrared wavelength is absorbed by the atmosphere and mainly causes temperature increase. | 28.1 | **76.4 \*\*\*** |
| The greenhouse effect is caused by naturally occurring gases in the atmosphere. | 68.5 | **82.0 \*\*** |
| **Causes and Effects of Climate Change** | | |
| Anthropogenic causes of global climate change (overall score) | 60.7 | **74.1 \*\*\*** |
| Fossil fuel combustion is a cause of climate change. | 93.8 | **98.6 \*\*\*** |
| The ozone hole in the upper atmosphere is not a cause of climate change. | 22.5 | **46.6 \*\*\*** |
| Real and possible consequences of global climate change (overall score) | 71.4 | **80.7 \*\*\*** |
| **Climate Change Mitigation Strategies** | | |
| Actions that will help reduce or slow down climate change (overall score) | 68.2 | **76.2 \*\*\*** |
| **Student understanding of the magnitude and causes of climate change: % of students who . . .** | | |
| Selected **climate change** [2] as the most important environmental problem facing the U.S. today | 21.3 | 60.7 [NA] |
| Selected **burning fossil fuels** [3] as the most significant cause of global climate change. | 80.9 | 98.9 [NA] |

[1] Bold values indicate statistically significant pre/post difference, at: \*\* $p < 0.01$, \*\*\* $p < 0.001$ (Wilcoxon signed-rank test). [2] Other choices included rain forest loss, loss of species to extinction, toxic waste, overpopulation, air pollution, water pollution, ozone depletion, destruction of ecosystems, acid rain, and urban sprawl. [3] Other choices included radioactive waste, livestock production, the ozone hole in the upper atmosphere, dumping trash into our oceans, waste rotting in landfills, destruction of tropical rainforest, nuclear power generation, and agricultural use of chemical fertilizer. [NA] No statistical comparisons were made between pre-test and post-test responses for these items.

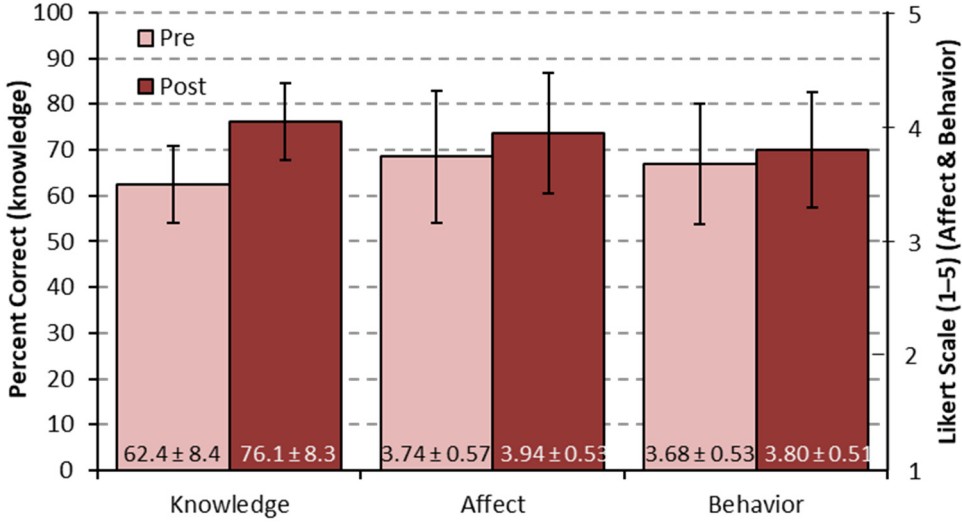

**Figure 3.** Student gains on climate literacy: knowledge, affect, and behavior. Gains are significant for knowledge ($p \ll 0.001$), affect ($p \ll 0.001$), and behavior ($p = 0.002$).

As expected (e.g., [33]), there was a significant correlation between students' knowledge, affect, and behavior pre-scores ($p < 0.01$). Also as expected (e.g., [34]), the correlation between affect and behavior ($\rho = 0.539$) was stronger than the correlation between either of these two subscales and knowledge scores ($\rho = 0.460$ and $0.0336$, respectively).

Tables 3 and 4 illustrate the types of overall gains made by students on the *Climate Literacy Questionnaire*. Students demonstrated significant pre/post improvement on 35 of the 42 cognitive items (examples in Table 3) and felt much more confident in their knowledge of climate change following the course (Table 4), with 86% responding that they know "a lot or quite a bit" about climate change on the post-test, compared to the pre-test response of 26%. The most remarkable gains were in the topic of climate science (Table 3). After taking

the climate change course, students better understood the abundance and importance of various greenhouse gases and the various mechanisms involved in the greenhouse effect that contribute to global warming. Most items that did not show a significant gain in this topic were already understood by students before taking the course—for example, 89% of the students already recognized the difference between weather and climate and 92% understood the relationship between the greenhouse effect and global warming.

**Table 4.** Student post scores on selected affective items.

| Survey Items (Followed by Likert-Type Response Option) | Average Score (out of 5) [1] | % Positive Response [2] |
|---|---|---|
| I feel I know (a lot, quite a bit) about global climate change. | **3.92 $\pm$ 0.51 \*\*\*** | 85.4 |
| I am (completely, mostly) convinced that global warming is happening. | **4.69 $\pm$ 0.47 \*\*\*** | 100.0 |
| Global warming is caused (mostly by human activities). | **4.72 $\pm$ 0.67 \*\*** | 80.9 |
| Global warming is an (urgent/very serious) threat to: | | |
| •     People in other countries | **4.08 $\pm$ 0.76 \*\*\*** | 79.8 |
| •     People in the United States | **3.63 $\pm$ 0.86 \*** | 51.7 |
| •     My family | 3.02 $\pm$ 1.00 | 28.1 |
| I would (strongly/somewhat favor) requiring automakers to increase the fuel efficiency of cars to 35 miles per gallon, even if a new car would cost $500 more. | **4.52 $\pm$ 0.83 \*\*** | 87.6 |
| I would (strongly/somewhat favor) increasing taxes on gasoline so people either drive less or buy cars that use less gas. | 3.24 $\pm$ 1.41 | 52.8 |

[1] Bold values indicate statistically significant pre/post difference, at: \* $p < 0.05$, \*\* $p < 0.01$, \*\*\* $p < 0.001$. [2] Positive response is described by parenthetical explanation for each item.

Differences between student responses on the pre- and post-questionnaire indicate that students were generally much more cognizant and concerned with the magnitude, causes, and effects of climate change following their participation in the course. When asked to identify the most important environmental problem facing the United States today, 61% of the students identified climate change on the post-test compared to only 21% on the pre-test (Table 3). They also displayed increased recognition of the link between climate change and fossil fuel combustion, as well as other anthropogenic causes such as livestock production and agricultural use of chemical fertilizers, and increased understanding that nuclear power and disposal of radioactive waste do not cause climate change. The number of students who were confused by the misconception that climate change is caused by the ozone hole in the upper atmosphere decreased after participating in the climate change course, although even on the post-test, 47% incorrectly identified this as a cause of global climate change. In terms of climate change mitigation, there was significant improvement in students' ability to correctly categorize three of nine strategies that would or would not help reduce global climate change.

Responses to affective and behavior questions indicate that course participation strengthened students' concern about climate change and its impacts and their willingness to participate in solutions (Table 4). At the end of the course, 100% of students responded that they were completely or mostly convinced that global warming is happening and 81% agreed that it is caused mostly by human activities. Most students felt global warming is a threat to people in other countries (80%), although far fewer were concerned about a threat to people closer to home, such as their local community or their family (30%). Our students generally favored increasing the price of commodities such as cars or household energy in order to support the implementation of energy efficiency and renewable energy technologies, while increasing taxes to accomplish similar goals was far less popular.

The course also had some benefit in improving 4 of 13 items within the behavior subscale. For example, after taking the climate change course, 96% of the students reported that they almost always or quite frequently recycle, 71% use previously-used or reusable shopping bags, 55% walk or ride a bike instead of driving short distances, and 92% turn off lights when they leave a room.

These indirect measures such as students' self-reported attributes on a questionnaire are subject to a range of limitations, including potential inconsistencies between what people 'say' and what they 'do' as well as the inability to collect in-depth explanations regarding the reasoning behind responses to the simplified Likert-type scale. While the latter problem is a clear limitation of this questionnaire-based study, the use of overall pre-post differences helped to minimize the moderating effects of self-reported answers, as these effects were likely to persist on both pre- and post-questionnaires.

### 4.2. Student Feelings of Competency Related to Climate Change and Engineering Goals

The self-efficacy items on the climate literacy questionnaire provided a means of assessing research question 3: At the end of the semester, do the students have the self-efficacy to effectively integrate climate change perspectives into their profession?

The climate literacy self-efficacy results show that students had a relatively high climate self-efficacy at the beginning of the class ($4.01 \pm 0.62$ out of 5), with statistically insignificant gains up to $4.07 \pm 0.65$ (post). The highest post scores were for questions related to the impact of actions by the US on global warming (91% agreed or strongly agreed) and believing that the students as individuals can take action to have a positive impact in both their personal (85%) and professional (87%) lives. The only significant change was an increase to 69% in the students' sense of urgency to take immediate and drastic action to reduce global warming and its associated major disruptions ($p = 0.0006$).

### 4.3. Variability in Climate Literacy and Self-Efficacy among Student Groups

Additional statistical analyses provided insight into the nature of climate literacy among students in this class. Potential differences in student outcomes were initially investigated using a series of Kruskal–Wallis one-way analyses of variance.

Three variables had an impact on student outcomes: gender, previous exposure to climate change topics in an earlier class, and student major. These three variables, further analyzed with stepwise multivariate regression analyses, were not independent. There was a strong correlation between gender and major ($\rho = 0.338$, $p < 0.01$) (Figure 2). Students in Group 2 majors and female students were also more likely to have had a prior course that included climate change (females, $\rho = 0.257$, $p < 0.05$; Group 2 majors, $\rho = 0.332$, $p < 0.05$).

Multivariate regression analyses revealed that previous exposure to climate change topics was the greatest predictor of student pre-scores on the affect and behavior subscales, followed closely by student gender ($p \leq 0.05$). Specifically, students with prior classes indicated that they were more strongly convinced that climate change is happening and responded more positively to questions about their level of concern about climate change and the degree to which they are adopting behaviors such as walking/biking, turning off appliances, and recycling/reusing materials (all differences significant at $p \leq 0.05$). This correlation with prior exposure to climate change in another class could be biased by fundamental and pre-existing environmental values that contributed to these students choosing classes that include climate change concepts, or it could be a function of the value of education in changing attitudes and behaviors through increased understanding and awareness of problems and an increased personal motivation.

While both male and female students had significant gains on the knowledge scale (Figure 4), the female students did not show significant gains on the affect or behavior scales because their pre scores were already high. The post scores for all subscales exceed the target (70% right or score >3.8 on affective and behavioral subscales) except for the self-reported behavior of male students. Identical conclusions were drawn for analysis of Group 1 (MAE, EE, and ChE) versus Group 2 majors (EnvE, CE, E&M), where accreditation requirements differ in terms of expectations for social and sustainability learning outcomes. With the strong correlation between gender and major (Figure 1), we were unable to identify which variable contributed more greatly to climate literacy.

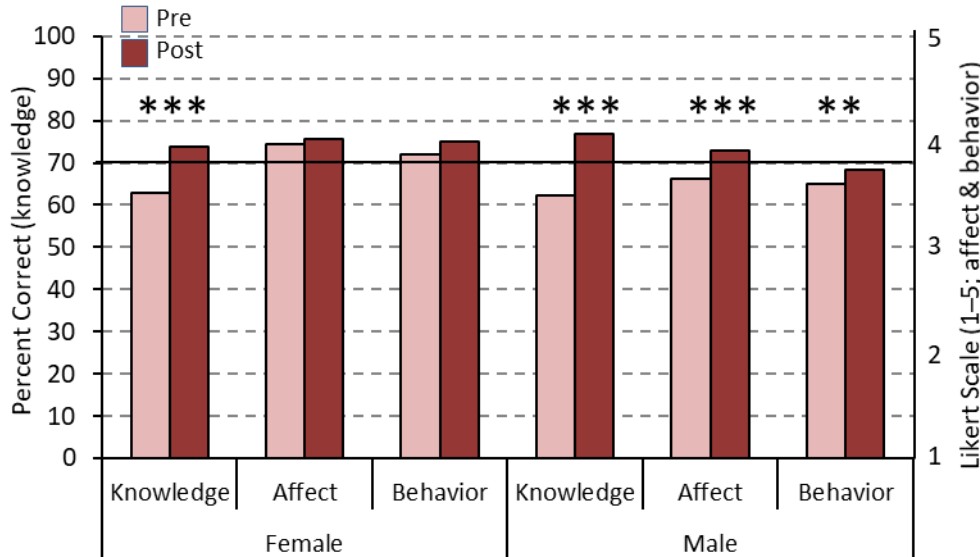

**Figure 4.** Variability in climate literacy scores by gender. ** = $p \leq 0.01$ and *** = $p \leq 0.001$.

There is some literature evidence to corroborate the differences we observed among majors. Meyers and Mertz [35] found that civil engineering students are more motivated to make the world a better place than mechanical engineering students (CE: 44.4% vs. ME: 40.7%, not statistically significant), whereas ME students are more interested in innovation and creativity (44.4%) compared with civil engineering students (5.6%). Goodwin et al. [36] found that engineering students generally do not identify with "global agency", which includes concepts such as "science and technologies will provide greater opportunities for future generations". In contrast, students in our study responded very positively to a similar question, "A career in science or engineering will help me contribute to solving the global climate problems", with high initial scores (4.23/5) and non-significant increases in the post-survey (4.30/5). Scores for women on this question were statistically higher than men for the pre ($p = 0.026$) survey and marginally higher on the post ($p = 0.087$) survey. Although women seem to be more pre-disposed to an attitude that engineering is a means to help solve global social and environmental problems, the men in this class were more strongly impacted in terms of improved climate self-efficacy and affect scores. By the end of the class, differences between men and women and among majors were reduced compared to the start of the class. Thus, providing a class like this appears to bring students to a similarly high level of attitude and self-efficacy required to contribute to solutions.

Student outcomes were not impacted by their grade level or year in which they took the class, although there were a few notable variations in student responses on the pre-questionnaire. For example, seniors began the course with higher average knowledge scores than juniors, as would be expected. Looking at changes over time, there were no variations in students' initial affect and behavior responses, and the only noteworthy trend in knowledge is that students who took the course in the later years had a greater understanding of climate change impacts and the disconnect between nuclear power and climate change compared to students in earlier years (data not shown). This general lack of change in students' climate knowledge over time may seem surprising given the recent surge in climate-related news from a variety of media sources. It is likely that, were we to repeat the study, we would find different results.

### 4.4. Competencies Required to Address Climate Change

While self-efficacy survey questions allow students to self-report what they believe they can do or accomplish, the review of their semester reports provides a direct assessment of their capabilities as an engineer to address climate change challenges. Review of the 30 reports completed over four years showed that 80% demonstrated at least proficient

(≥4.0) problem-solving and critical-thinking competencies (Figure 5). As a whole, the average among reports met the target "proficient" score in two of four attributes, with the greatest strength in formulating and identifying a climate-change-related problem and weakest in appropriately using data and evidence.

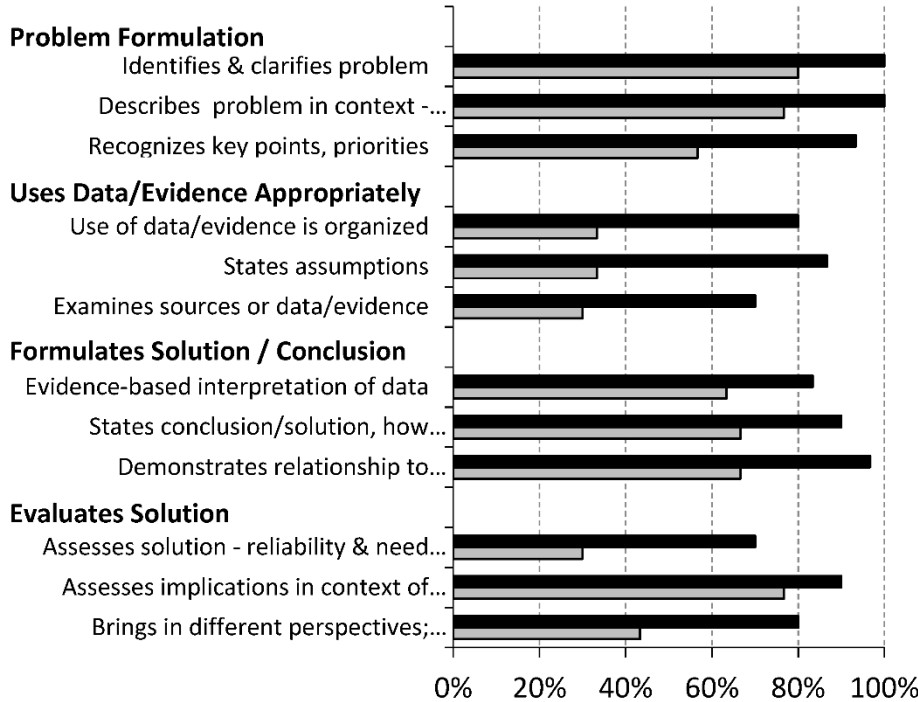

**Figure 5.** Percent of reports from all years reaching high scores on the competencies rubric. Black bars represent the percentage of reports achieving at least proficient (≥4); gray bars are the percent achieving a superior score of 5. See Table 2 for details about these competencies.

Detailed results from the competencies rubric can be interpreted to assess the percentage of reports that met the proficient or superior scores in each of the 12 competency expectations (Figure 4). These results are very encouraging as they show that engineers are strong in the critically important climate change competencies related to considering multidisciplinary perspectives and broader systems thinking (high scores for the "Describes problem in context" (100% of reports scored at least proficient (≥4)) and "Assess implications in context" competencies (97% at least proficient)). The reports showed lower levels of proficiency in students' ability to critically analyze the data, evidence, and assumptions that they used to address climate change challenges. In many cases, the climate modeling tools and data used in their projects were previously used in class activities and homework. Thus, it is likely that they had a pre-conceived expectation that these were excellent resources without explicitly evaluating or discussing that in their reports.

*4.5. Limitations and Future Directions*

This research provides a case study view of a single class that used an active learning and engagement pedagogical approach to teach engineering students about climate science and solutions. It clearly shows the value of this approach for increasing all dimensions of climate literacy and the students' ability to apply that literacy to a semester project that required critical-thinking skills. However, the study lacked control groups to assess how climate literacy and critical thinking capabilities change for other students not in a climate change class or not in one with this particular pedagogical approach. Additional research that included controls or a longitudinal assessment of the lasting impact of the class would add greater insight.

As we write this paper in the summer of 2021, record-breaking 110 + °F (43 °C) records set in the northwest regions of the United States and Greece and horrific flooding in China and Germany are raising the visibility of our changing climate. Expanding the study to assess the general pre-class climate literacy of engineering students after these events and increased news reporting would be an interesting follow-up study to address how this is impacting our engineering students especially across majors and genders.

## 5. Conclusions

The overall assessment presented here shows that students in a climate change class designed for engineers showed substantially improved climate literacy, higher climate-related and professional self-efficacy, and proficiency in their problem-solving and critical-thinking skills related to mitigation or adaptation solutions. When challenged to do so, engineering students can broaden their perspectives and integrate at least an awareness of the social and environmental impacts of climate change and its potential mitigation and adaptation solutions. The acquisition of these skills as a student bodes well for the engineering profession in two key ways:

(1) Students are better prepared for multidimensional and systems thinking for any grand challenges they may face, not just climate change; and,
(2) These students are better prepared to integrate climate change and other environmental and social impacts into their engineering decisions, which should set the stage for developing engineering professionals that are needed for leadership to address global societal challenges.

**Author Contributions: Conceptualization:** S.E.P., J.E.D. and S.D.; Methodology: S.E.P., J.E.D. and S.D.; formal analysis: J.E.D. and S.E.P.; resources, S.E.P. and S.D.; writing—original draft preparation, S.E.P. and J.E.D.; writing—review and editing, S.E.P., J.E.D. and S.D.; visualization, S.E.P.; project administration, S.E.P.; funding acquisition, S.E.P., J.E.D. and S.D. All authors have read and agreed to the published version of the manuscript.

**Funding:** This research was initially supported by the National Aeronautics and Space Administration (NASA) NICE (NASA Innovations in Climate Education) program (NNX10AB57A). The findings and opinions presented here do not necessarily reflect the opinions of the funding agency.

**Institutional Review Board Statement:** The protocols were reviewed and approved by Clarkson University's Institutional Review Board (IRB # 10-24).

**Informed Consent Statement:** Informed consent was obtained from all subjects involved in the study.

**Data Availability Statement:** Data are available through Mendalay, available online: https://data.mendeley.com/datasets/2wv825np5m/1 (accessed on 28 June 2021).

**Conflicts of Interest:** The authors declare no conflict of interest.

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
