# Peer review of "Climate Literacy—Imperative Competencies for Tomorrow’s Engineers"

_sustainability, doi:10.3390/su13179684_

Round 1
Reviewer 1 Report
This paper shows the pedagogical approach used in a class for engineering students that helped contribute to significant gains in engineering students’ climate literacy and critical thinking competencies. The paper reads well and can be accepted after minor revisions.
See the below:
- Lines 233-245: What is your aim to make two groups? Please make clear your intention in accordance with your hypothesis.
- Line 256-267: Please make clearer the scale of knowledge and also the structure of affective, behavioral, cognitive aspects and knowledge with the framework of pedagogy you based on.
- Section 3.5: Please make clear the statistical tests applied at anywhere in this section. In case of observing significant difference between pre and post data of the same sample, I think the paired t-test should be conducted.
- Section 4.3: It may be natural that the result of the multivariate regression analyses showed that the previous exposure to climate change topics was the greatest predictor because the other two explanatory variables had multicollinearity. I'm wondering if the results with two independent variables will be more convincing.
Reviewer 2 Report
This is a useful paper that demonstrates that engineering students' views on climate change can shift with appropriate instruction and active learning. This is a tribute to the teaching team, because students' attitudes to climate change can be difficult to shift, particularly for those who see themselves working in carbon intensive industries (coal, oil, gas). I was also astonished by the long list of resources you make available to the students (Table 1).
I liked the gender comparison in Figure 3. I was left wondering whether it would be useful to include a similar comparison for majors (groups 1 and 2) and also those who had taken a climate change course previously, versus those who had not. Maybe the results are similar to Figure 3, but they might be interesting. I'm assuming that the shifts for the latter group would be the biggest of all.
There are a couple of minor typos to fix: Questionnaires, line 252, and student, line 432. Otherwise, very careful writing! Well done.
Reviewer 3 Report
This is a well-organized and clearly presented analysis of valuable data, though it may not support the depth of conclusions you draw. The general topic of science education in the Anthropocene is perhaps one of the most critical issues facing us today, and (not being an engineer myself) engineering seems one of the most logical places to teach about it. Being a geographer, I confess my natural bias toward integrating more teaching on climate change into any and every discipline. So, I think this is a great project with immense potential for adaptation into other scientific fields.
My suggestions mostly concern the degree to which analysis of your survey data is adequately contextualized in broader (fluid) social, cultural, political conditions, both within academia and beyond. Some are more minor details, such as:
- In Section 3.2 (participants) – is it possible to also calculate the percent of students from each major/discipline who elected to take the course? (e.g. if 27% of all students in the course majored in environmental engineering, did that mean every single environmental engineer student signed up? Or just a handful of them?). This might be a useful descriptive measure to tease out the role that self-selection may play.
- Where you introduce the final project (lines 151 to 152) – without going into a lot of detail -- could you indicate if students were assigned topics, selected from a list, or had to come up with their own? (again, getting at the question of self-selection and following natural interests)
- Line 269 – can you clarify the same climate literacy questionnaire instrument was administered both pre and post-survey? (and also clarify that there was no attempt at pre-testing critical thinking, correct?)
- Starting around Line 341, when you get to the analysis of cognitive and affective survey results, from a social science perspective I’d recommend less reliance on statistical significance measures and more critical interpretation to account for both potential willful exaggerations (e.g. recycling behavior) or misidentifications (e.g. students who didn’t want to impose additional gasoline taxes – was it to keep gas prices cheaper, or because they only support political movements to abolish fossil fuels completely?)
- It seems safe to conclude that the climate change problem solving and critical thinking rubric assesses critical thinking skills successfully applied to a complex problem (of which climate change can be one of many examples), but I’m not sure the data supports drawing further conclusions about thinking critically about climate change in particular.
- I notice you carefully avoid saying it, but I get the impression the takeaway is that if student’s climate literacy is improved (e.g. factual knowledge, the “correct” affective answers offered), that will lead to enhanced awareness of how climate issues impact engineering as well as improved critical skills to address them. I don’t think this data supports that … yet … leading to my last few comments.
More broadly, there’s a flat empiricism to overall analysis I would encourage you to transcend (which also allows you to highlight some of the more innovative aspects of your work):
- In the introduction, you allude to Climate Change Education (CCE) as a homogeneous movement with agreed-upon goals, objectives, and methods. In reality, of course, it isn’t, which makes statements like (lines 84-88) “Critical among these skills is an awareness and respect for environmental and social issues at both local and global scales, transdisciplinary (or multidimensional) and systems thinking to move beyond defining solutions based only on technology and cost criteria, and an understanding of the complexities of our global grand challenges” oversimplify nuanced discourse and overlook points of epistemological fragmentation. Another example, line 186, from the curriculum: “Social science aspects of climate modeling - scenarios for future socio-economic futures.” How to address the multiple often conflicting interpretations and theoretical perspectives underlying this statement is equally important. I think even a brief engagement with these issues would strengthen your overall argument.
- There is a fundamental periodicity issue lurking. You mention this class dates back to 2010, that you have four years of survey data, and that the class has been offered every other year. I may have missed it, but I didn’t see specific details about the timing of data collection. If your data spans the entire 2010s decade, the last students surveyed literally inhabited a different world (environmentally, educationally, politically) than the first set of students. Even if it occurred over four consecutive years, that’s still a lot of change. You do suggest (line 387) there was no statistical difference in responses across years, but that too may bear further comment.
- Finally, there are a few unanswered questions that would make excellent ‘future directions’ in your conclusion. Is one semester-long course enough? Could you recommend additional qualitative measures to explore climate literacy? Does this course suffer the fate of many other ‘cultural diversity’ type courses, beloved for that semester but ultimately intellectually quarantined from the students’ required coursework? Longitudinal data revisiting these same students at the end of their degree programs, or even a few years into their engineering careers, would be fascinating.
I look forward to seeing a final version in print!
Reviewer 4 Report
This paper presented an interesting study of the effects of the knowledge increase / behaviourial changes in engineering students before and afgter taking a course in climate change. The results are not unexpected (anyone taking a course in a particular topic will undoubtably see their knowledge / attitude in this topic change). So there is little contribution to current knowledge in this aspect. What the paper does provide, is a clear case study and also numerical analysis from this particular study which could be useful for future researchers in this topic.
Round 2
Reviewer 4 Report
The previous comments have been adequately addresed.